# The return of the singularities:
# Applications of the smeared null energy condition

**Ben Freivogel [1], Eleni-Alexandra Kontou[1,2]⋆ and Dimitrios Krommydas [3]**

**1** ITFA and GRAPPA, Universiteit van Amsterdam, Science Park 904,
Amsterdam, The Netherlands
**2** Department of Physics, College of the Holy Cross,
Worcester, Massachusetts 01610, USA
**3** Instituut-Lorentz, Universiteit Leiden, P.O. Box 9506,
2300 RA Leiden, The Netherlands

⋆ e.a.kontou@uva.nl

## Abstract

The classic singularity theorems of General Relativity rely on energy conditions that can be violated in semiclassical gravity. Here, we provide motivation for an energy condition obeyed by semiclassical gravity: the smeared null energy condition (SNEC), a proposed bound on the weighted average of the null energy along a finite portion of a null geodesic. We then prove a semiclassical singularity theorem using SNEC as an assumption. This theorem extends the Penrose theorem to semiclassical gravity. We also apply our bound to evaporating black holes and the traversable wormhole of Maldacena–Milekhin–Popov, and comment on the relationship of our results to other proposed semiclassical singularity theorems.

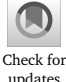

# 1  Introduction

General relativity allows for any spacetime geometry, even those with exotic features such as wormholes and causality violation. Given any metric, one can simply use the Einstein equation to find the appropriate stress-energy tensor. However, early on in the history of general relativity it was hypothesized that matter obeys certain restrictions called *energy conditions* (see [1] for a recent review). These energy conditions are bounds on the contracted stress energy tensor and they express properties expected of reasonable matter such as the positivity of the energy density. The classical energy conditions, which bound the energy density and similar quantities at every point in spacetime, were proven to be violated in all quantum field theories [2].

Bounds on the null-contracted stress tensor are particularly useful. The singularity theorem of Penrose [3] was proven by assuming the Null Energy Condition (NEC)

$$T_{kk} \geq 0\,, \tag{1}$$

where $T_{kk}$ is the stress tensor contracted with an arbitrary null vector. While this condition is satisfied by all sensible classical theories, all quantum field theories contain states in which $\langle T_{kk}(x^\mu)\rangle$ is negative.

Recently, two of us proposed that while the null energy at a point can be arbitrarily negative, the *average* of the null energy over a piece of a null geodesic is bounded below in semiclassical gravity [4]. Schematically, this smeared null energy condition (SNEC) claims that the average of the null energy over an achronal portion of a null geodesic with length $\tau$ is bounded by

$$\langle T_{kk}\rangle_\tau \gtrsim -\frac{1}{G_N \tau^2}\,. \tag{2}$$

More precisely, the claim is

$$\int_{-\infty}^{+\infty} d\lambda\, g^2(\lambda) \langle T_{kk}(x^\mu(\lambda))\rangle \geq -\frac{4B}{G_N} \int_{-\infty}^{+\infty} d\lambda \left(g'(\lambda)\right)^2\,, \tag{3}$$

where $x^\mu(\lambda)$ is a null geodesic, $g(\lambda)$ is a differentiable 'smearing function' that controls the region where the null energy is averaged, $B$ is a constant and $G_N$ is the Newton constant. The presence of $G_N$, ensures that (3) is relevant even in theories with a large number of species. When gravity is coupled to such theories, the renormalized $G_N$ to 1-loop order is $G_N \sim \frac{\ell_{UV}^{d-2}}{N}$. Since the stress tensor in theories with a number of fields $N$ is essentially $N$ times the stress tensor of each field, the $N$s cancel out to leave SNEC exactly as strong as in theories with fewer

fields. Note that the stress energy tensor here is contracted with the tangent vector to the null geodesic so the expression is invariant under reparametrization of the affine parameter.

We expect SNEC to be valid in the context of semi-classical gravity. Here, by semi-classical gravity, we mean a treatment of classical gravity coupled to quantum fields, where the metric is sourced by the expectation value of the stress tensor. This description can certainly be valid in quantum states with more than two particles, but it breaks down when the geometry is highly curved, or when the stress tensor has large quantum fluctuations on the characteristic distance scales relevant to the solution. A detailed understanding of the regime of validity of the semi-classical approximation is an interesting question, but not the focus of this work.

SNEC was proven, and the coefficient $B$ fixed, by Leichenauer and Levine within the framework of induced gravity on a brane [5].

The condition can be obtained from the one proposed in Ref. [4]

$$\int_{-\infty}^{\infty} d\lambda f(\lambda) T_{kk}(x^{\mu}(\lambda))\rangle \geq -\frac{B}{G_N} \int_{-\infty}^{+\infty} d\lambda \frac{(f'(\lambda))^2}{f(\lambda)}, \tag{4}$$

by defining $g(\lambda)^2 = f(\lambda)$.

In this paper, we

- Motivate the precise form of the bound shown above.

- Argue, by examining examples, that this bound may remain valid when smearing over distances of order the radius of curvature or larger. Explicit examples we consider are

  - Evaporating black holes.
  - The traversable wormhole of Maldacena, Milekhin, and Popov [6]
  - The counter-example to bounds of this form given by Fewster and Roman [7]

- Prove a singularity theorem, using SNEC as the assumption, which applies to real black holes.

In addition, in Appendix A we prove the (2d CFT) Quantum Energy Inequality of [8], in spacetimes globally conformal to Minkowski and for arbitrary smearing functions.

We begin in Sec. 2, where we provide motivation for SNEC using an argument of Wall [9]. We also show how a Lorentz invariant UV cutoff evades the Fewster-Roman no-go result for bounds of this form. In Sec. 3 we apply the bound go a variety of situations, starting with the ANEC limit and evaporating black holes. In Sec. 3.3 we closely examine the Maldacena–Milekhin–Popov wormhole and apply the SNEC bounds both in four and two dimensions. In Sec. 4 we prove a semiclassical singularity theorem using SNEC as an assumption and apply it to evaporating black holes, comparing our results to other approaches. We conclude in 5 with discussion and ideas for future work.

We work in units where $c = \hbar = 1$, use metric signature $(+,-,-,-)$ and assume $d$-spacetime dimensions unless otherwise stated.

**Relation to Previous Work.** Quantum field theories often obey energy conditions averaged over an entire geodesic and quantum energy inequalities (QEIs), lower bounds on the average renormalized stress-energy tensor in some region. The latter were introduced by Ford [10] and since then have been derived for scalar [11] and fermionic fields [12, 13] in flat and curved spacetimes [14]. In terms of interacting fields little progress has been made with concrete examples only in two dimensions [15].

**ANEC.** The previous examples of QEIs are all for averages over a timelike curve. The averaged null energy condition (ANEC)

$$\int_\gamma d\lambda \, \langle T_{kk} \rangle \geq 0 \,, \tag{5}$$

where $\langle T_{kk} \rangle$ is the expectation value of the null contracted stress-energy tensor and $\gamma$ is a complete null geodesic is the most common example of a null averaged energy condition. It is believed that the self-consistent achronal ANEC is a fundamental property of physical matter at least at the semiclassical context. All known violations of the self-consistent achronal ANEC involve Planck length distances [16], outside the range of validity of the semiclassical approximation. There are numerous examples of proofs of the achronal ANEC. To mention some, Kontou and Olum [17] proved the achronal ANEC for free scalar fields in curved spacetimes while Flanagan and Wald [18] provided the first proof of ANEC in Miskowski space that includes backreaction up to second order in perturbation theory. In a different approach, Wall [19] derived ANEC from the generalized second law of thermodynamics and more recently Faulkner et al. [20] derived ANEC using modular Hamiltonians for general fields. Using similar methods Rosso proved ANEC for general quantum field theories in the near horizon geometry of spherical extremal black holes [21].

**QNEC.** Our energy condition and resulting singularity theorem differ in nature from the beautiful results based on the generalized entropy. Bousso et al. introduced the quantum null energy condition (QNEC) [22]. This is a lower bound on $\langle T_{kk} \rangle$ at a single point $p$; the bound is computed from the von Neumann entropy, in a region $\Sigma$ whose boundary contains the point $p$. In particular

$$\langle T_{kk} \rangle \geq \frac{1}{2\pi a} S''[\Sigma] \,, \tag{6}$$

where $S''$ is a second functional derivative with respect to deformations of $\Sigma$ in the null direction $k_a$ at $p$, and $a$ is the transverse area element. QNEC arises from the quantum focusing conjecture [22] and it was proven for relativistic QFTs in $d \geq 2$-dimensional Minkowski spacetime [23]. A non-exhaustive list of other proofs of QNEC can be found in [24–26] and references therein. QNEC is a bound on a local quantity, at the cost of introducing a somewhat complicated state-dependent quantity on the right hand side of the inequality.

Our bound is simpler, in the sense that the smeared stress tensor is bounded by a c-number, while in the case of the QNEC, the quantity appearing on the right side is the derivative of the entanglement entropy, which depends on the quantum state. As a result, we can prove simpler singularity theorems: our singularity theorem says that an initial surface with sufficiently negative contraction $\theta$ implies the existence of a singularity, whereas the singularity theorems in the generalized entropy approach refer to the generalized contraction $\Theta$, which to our knowledge is not an observable.

Another important difference between QNEC and SNEC is that the operator appearing in SNEC involves averaging the stress tensor with a smearing function. If the smearing function is not smooth enough, the bound diverges and the operator is unbounded. In contrast, the integrated version of the QNEC places a bound in the case that the smearing function is a top-hat function; in this case our bound diverges. So the bottom line is that the smeared operators we consider are rather different from the quantity that is bounded by QNEC, even though they appear similar at first glance.

However, our simpler theorem also comes at a cost: for an evaporating Schwarzschild black hole, our singularity theorem is weaker, in the sense that one must go deeper inside the horizon to find a sufficiently trapped surface. The generalized entropy approach allows one to deduce a singularity theorem at the location of the quantum extremal surface, which is

typically only an ultraviolet length scale inside the black hole, while our theorem requires us to go a fraction of the Schwarzschild radius inside.

**Null QEIs**    But what about a null-averaged QEI? Flanagan provided the first example in two-dimensions in flat [27] and curved spacetimes [28] while Fewster and Hollands [8] derived a similar bound for classes of interacting conformal field theories (CFTs). Unlike the timelike case it is not straightforward to generalize these results in four dimensions. In fact Fewster and Roman [8] argued using an explicit counterexample that weighted averages of the null-contracted stress-energy tensor along null geodesics in four dimensions are unbounded from below. In [8] the authors considered a sequence of vacuum–plus–two–particle states. The three-momenta of the excited modes become more and more parallel to the spatial part of the null vector that the stress-energy tensor was contracted with making the lower bound of the energy density divergent. The introduction of SNEC [4] overcame that problem by introducing a UV momentum cutoff.

**Applications.**    Energy conditions have several interesting applications. One of the most important is restrictions on exotic spacetimes, for example those with wormholes. It was shown [29] that achronal ANEC is sufficient to rule out so called "short" wormholes. In those it takes longer to travel through the ambient space than the wormhole, creating a shortcut in spacetime and thus allowing for causality violations. However, "long" wormholes, those that it takes longer to travel through the throat than outside are allowed. In the past few years there are examples for those kind of wormholes with perhaps the most famous being the one proposed by Maldacena, Milekhin and Popov [6]. Even though achronal ANEC is obeyed in this situation it was not clear if a stronger bound like SNEC is also obeyed and whether that bound is saturated or not. We answer that question with bounds in both four and two dimensions.

A second major application of energy conditions is singularity theorems, which predict the formation of a singularity, defined in that context as the spacetime possessing at least one incomplete causal geodesic. The singularity theorems of Penrose [3] and Hawking [30] were the first to predict a singularity under general assumptions without restricting to symmetric spacetimes. Senovilla [31] has described the skeleton of singularity theorems as a 'pattern theorem' with three assumptions. An initial or boundary condition which establishes the initial focusing of a congruence of geodesics, an energy condition which ensures that the focusing effect continues, and a causality condition which removes the possibility of closed timelike curves. The contradiction of the geodesic focusing with the causality condition and their length extremization property leads to geodesic incompleteness. We will divide singularity theorems into 'Hawking-type' and 'Penrose-type', depending on whether they demonstrate timelike or null geodesic incompleteness respectively.

The original singularity theorems used pointwise energy conditions such as the NEC eq.(1). To prove a singularity theorem semiclassically, it is necessary to have an energy condition obeyed by quantum fields. Early work generalized Penrose's theorem using averaged energy conditions [32–34]. Fewster and Galloway [35] presented proofs of singularity theorems with conditions inspired by QEIs. More recently Fewster and Kontou [36] proved singularity theorems with similar conditions using index form methods. Additionally they estimated the required initial contraction on a Cauchy surface to guarantee the formation of a focal point both in the case of timelike and null geodesics. However, their conditions are still not derived by a QFT. In the timelike case the relevant condition is the quantum strong energy inequality, derived for the quantum scalar field by Fewster and Kontou [37]. The same authors use this inequality to derive the first semiclassical singularity theorem in the timelike case [38]. In this work we focus on null geodesics where the relevant condition is SNEC.

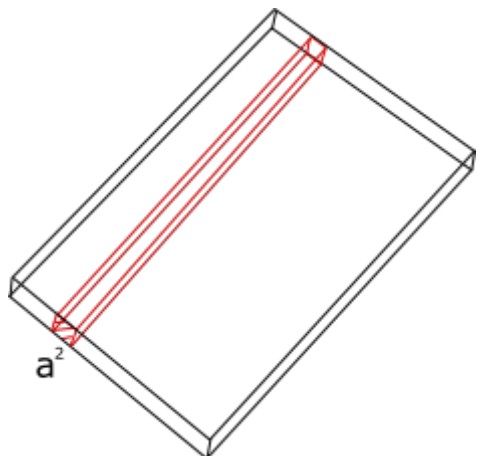

Figure 1: Schematic representation of pencils on the lightsheet.

## 2 Motivation for SNEC

In this section we provide some motivation for SNEC, additional to that given in [4] and [5]. First we sketch a proof for the QFT version of SNEC in arbitrary dimensional Minkowski space-time. Then we revisit the Fewster-Roman counterexample to show that the Lorentz invariant SNEC bound is obeyed in the class of vacuum plus 2-particle states.

### 2.1 Derivation of SNEC using 'pencils'

A nice motivation for the field theory version of SNEC comes from an argument due to Jackson Fliss. Here we give a quick overview since it motivates the precise version of the bound we will use; details and subtleties appear in [39].

Wall showed [9] that free field theory in higher-dimensional Minkowski space factorizes on the lightsheet into a collection of 2d theories living on 'pencils.' More precisely, the null plane is divided into pencils with area $a$. The pencils are a null line regularized by a $(d-2)$-dimensional transversal area as shown in Fig. 1. Effectively, they are only one-dimensional objects. We assume that $a$ is the smallest scale in the problem; modes with wavelength shorter than $a^{1/(d-2)}$ cannot be excited.

The precise constants in the bound we motivate here will depend on the details of the cutoff scheme, so we do not keep track of order one constants in this section. The theory on each pencil is a chiral 2d CFT with central charge proportional to the number of fields in the higher dimensional theory; for $N$ free scalars, the relation is

$$c = N. \tag{7}$$

The higher dimensional stress tensor is then related to the 2d stress tensor by

$$T_{++} = \frac{1}{a} T_{++}^{(2)}, \tag{8}$$

where the equality is up to order one numbers. The SNEC quantity is then related to a 2d CFT quantity

$$\int_{-\infty}^{+\infty} d\lambda\, g^2(\lambda) \langle T_{++}(x^\mu(\lambda)) \rangle = \frac{1}{a} \int_{-\infty}^{+\infty} d\lambda\, g^2(\lambda) \langle T_{++}^{(2)}(x^\mu(\lambda)) \rangle. \tag{9}$$

Now we can use the 2d CFT results of [8] (see also Appendix A)

$$\int_{-\infty}^{+\infty} d\lambda\, g^2(\lambda) \langle T_{++}^{(2)}(x^\mu(\lambda)) \rangle \geq -\frac{c}{12\pi} \int g'(\lambda)^2 d\lambda, \tag{10}$$

(where we have used $g^2 \equiv H$ in relation to [8]) to obtain

$$\int_{-\infty}^{+\infty} d\lambda\, g^2(\lambda) \langle T_{++}(x^\mu(\lambda)) \rangle \geq -\frac{N}{12\pi a} \int g'(\lambda)^2 d\lambda\,. \tag{11}$$

Since $a^{1/(d-2)}$ is the smallest allowed wavelength, it should be identified with the UV cutoff.

As explained in [4], using the lore relating the UV cutoff to the Planck length,

$$G_N \lesssim \frac{\ell_{\mathrm{UV}}^{d-2}}{N} \lesssim \frac{a}{N}\,. \tag{12}$$

The above equation implies the gravitational form of our bound,

$$\int d\lambda\, g^2(\lambda) \langle T_{++}(x^\mu(\lambda)) \rangle \geq -\frac{4B}{G_N} \int d\lambda\, g'(\lambda)^2\,. \tag{13}$$

Note that this argument is not a general proof, since the decomposition into pencils works only in Minkowski spacetime for free or super-renormalizable theories [22]. However, it does motivate the particular form of the right hand side. Other possible quantities could appear on the right hand side of the bound, including higher derivatives of the function $g$; we will focus on this proposal since it is the correct form for the cases where the pencil decomposition is valid. The extension of the proof to interacting QFTs is beyond the scope of this work but it could be possible using methods developed in [40] and [41].

## 2.2 The physical meaning of $B$

As we showed in the previous section, when we consider free fields on Minkowski spacetime we have Eq. (11) the undefined overall constant is an order one number. The constant $B$ in this case arises from the relation between the UV cutoff and the Planck length. To have $B$ be an order one number we need to saturate the inequality

$$N G_N \lesssim \ell_{\mathrm{UV}}^{d-2}\,. \tag{14}$$

This was the case for the induced gravity proof of [5] where they derived $B = 1/32\pi$.

However, it is well motivated to consider a $B \ll 1$ as (14) is typically not saturated in controlled constructions. For example, in controlled string theory constructions the string scale and the Kaluza-Klein scale are both well below the Planck scale, and these scales instead set the UV cutoff of the theory.

Additionally, a controlled construction in an EFT with cutoff $\ell_{\mathrm{UV}}$ should not excite modes near the UV cutoff, and so should not saturate the field theory SNEC, and therefore cannot saturate the gravity SNEC. To summarize: when semi-classical gravity is well under control, $B \ll 1$.

Since SNEC is only proven for free fields in Minkowski we can use an undefined $B$ in order to have an inequality valid for other fields and curvature. In these cases we can consider $B$ as the smallest possible number in order to have a generally obeyed bound.

## 2.3 Fewster-Roman counterexample

Any proposed bound must first address the argument of Fewster and Roman [7]. They constructed an explicit family of states in free quantum field theory that have arbitrarily negative values for the smeared null energy.

This counter-example is given in quantum field theory without gravity, so it is appropriate to compare their construction to the field theory version of our bound,

$$\int_{-\infty}^{+\infty} d\lambda\, g^2(\lambda) \langle T_{++}(x^\mu(\lambda)) \rangle \geq -\# \frac{N}{\ell_{\text{UV}}^{d-2}} \int g'(\lambda)^2 d\lambda\,. \tag{15}$$

Since the UV cutoff appears explicitly, the number $\#$ on the right side depends on the precise cutoff scheme. However, it is still well-defined to ask whether a bound of this form holds. (It is also interesting to look for bounds that survive the limit $\ell_{\text{UV}} \to 0$; we hope to return to this in the future.)

In this subsection, we show that when a UV cutoff is imposed on the states of Fewster and Roman, our bound eq.(15) is respected. This point was also addressed in [4], but there the UV cutoff was imposed in a way that was not manifestly Lorentz-invariant[1]. Here we improve the analysis by imposing a manifestly Lorentz-invariant UV cutoff.

The counterexample of Fewster and Roman makes use of a particular class of states, which are a superposition of the vacuum and 2-particle states,

$$|\psi\rangle = N_\alpha \left( 1 + \int \frac{d^3\mathbf{k}}{(2\pi)^3} \frac{d^3\mathbf{k}'}{(2\pi)^3} b_\alpha(\mathbf{k}, \mathbf{k}') a_{\mathbf{k}}^\dagger a_{\mathbf{k}'}^\dagger \right) |0\rangle\,. \tag{16}$$

Here the momenta of the particles appearing in the 2-particle state are controlled by the function $b_\alpha(\mathbf{k}, \mathbf{k}')$. Fewster and Roman consider a one-parameter family of states labelled by the parameter $\alpha$.

They consider the limit $\alpha \to 0$, and show that the smeared null energy diverges in this limit. Concretely, for a particular family of functions $b_\alpha$, the smeared null energy takes the form

$$< T_{++}^s >_\alpha = -\alpha^{\sigma - 2(\nu+1)} A, \tag{17}$$

where $\alpha \ll 1$ labels a class of Hadamard states whose negative energy density diverges as $\alpha \to 0$, while $A$ captures all the $\alpha$-independent information. Finally, $\sigma$ and $\nu$ are two numbers that parameterize the state, which must obey

$$\sigma > 2\nu + 3/2\,. \tag{18}$$

(This is a rewriting of equations (II.22) and (II.23) of [7].) Therefore, in the $\alpha \to 0$ limit, the smeared null energy is

$$\langle T_{++}^s \rangle_\alpha = -A\, \alpha^{\gamma - 1/2}\,, \tag{19}$$

with $\gamma > 0$. This diverges as $\alpha \to 0$ as long as $\gamma < 1/2$. Because the smearing function does not depend on $\alpha$, the right hand side of SNEC is independent of $\alpha$, so this family of states appears to violate SNEC.

We show now that imposing a UV cutoff regulates this divergence so that our bound is respected. The idea of the Fewster-Roman counterexample is to 'boost' the momenta $\mathbf{k}, \mathbf{k}'$ along the direction of the light ray of interest. The 'unboosted' state (defined by $\alpha = 1$) has maximum momentum

$$|\mathbf{k}|_{\max} = \Lambda_0\,. \tag{20}$$

Following [7], we refer to $\Lambda_0$ as the center of mass momentum.

As the state is boosted, $\alpha \to 0$, the momenta grow as $|\mathbf{k}| \sim \Lambda_0/\alpha$. The angle between the 3-momentum $\mathbf{k}$ and the null vector is given by [7]

$$\cos\theta = 1 - \alpha\,, \tag{21}$$

---

[1]We thank Ken Olum and Alex Vilenkin for discussions on this point.

so that as $\alpha \to 0$, the momenta become nearly parallel with the null direction.

Once we have chosen a null ray, the magnitude of the transverse momentum $k_\perp$ is invariant under Lorentz transformations that preserve the null ray. Therefore, it is Lorentz invariant to impose a cutoff on the transverse momentum,

$$|k_\perp| \leq \frac{1}{\ell_{\mathrm{UV}}}. \tag{22}$$

In [4], the cut-off was imposed on the full 3-momentum $\mathbf{k} = \Lambda_0/\alpha$. Here, we restrict only the *transverse* directions of the momentum, $k_\perp$.

As $\alpha \to 0$, the momentum and also the negative part of the energy density go to infinity. Therefore, we are interested in small $\alpha$. Expanding eq.(21), we find

$$\sqrt{\alpha} \sim \theta , \tag{23}$$

while the transverse component of the momentum is

$$k_\perp = |\mathbf{k}| \sin \theta \approx |\mathbf{k}| \theta \sim |\mathbf{k}| \sqrt{\alpha}. \tag{24}$$

Recalling that $|\mathbf{k}| \approx \Lambda_0/\alpha$, this becomes

$$k_\perp \sim \frac{\Lambda_0}{\alpha} \cdot \sqrt{\alpha} = \frac{\Lambda_0}{\sqrt{\alpha}}. \tag{25}$$

In [4], we required $k = \frac{\Lambda_0}{\alpha} < \ell_{UV}^{-1}$, but now that we impose a cut-off only on the transverse momenta, we obtain the relaxed condition

$$k_\perp = \frac{\Lambda_0}{\sqrt{\alpha}} < \ell_{UV}^{-1}. \tag{26}$$

So finally, our UV cutoff imposes a cutoff on the boost parameter $\alpha$,

$$\sqrt{\alpha} > \Lambda_0 \ell_{\mathrm{UV}}. \tag{27}$$

Plugging this into equation eq.(19) for the smeared null energy, we obtain

$$< T_{++}^s >_\alpha = -\alpha^{\gamma-1/2} A \geq -\frac{1}{\Lambda_0 \ell_{\mathrm{UV}}} A. \tag{28}$$

Recall that $A$ and $\Lambda_0$ depend only on the unboosted state. Therefore, boosting leads to a divergence of $\ell_{\mathrm{UV}}^{-1}$, which is milder than the SNEC bound, which is proportional to $\ell_{\mathrm{UV}}^{-2}$. This shows that the Fewster-Roman technique of boosting does not lead to a violation of SNEC. One would still like to *prove* SNEC, including order one factors, in the theories considered by [7].

**Frame independent cutoff.** As an aside, note that one can derive the same condition in a way that is manifestly invariant under all Lorentz transformations, not just those that preserve the light ray of interest. Instead of bounding $k_\perp$, we impose that the center of mass energy between any two momenta is smaller than the cutoff.

We start from the center of mass energy

$$E_{cm}^2 = \omega \cdot \omega' - \mathbf{k} \cdot \mathbf{k}'. \tag{29}$$

Remembering that $\theta$, the angle between $\mathbf{k}$ and $\mathbf{k}'$ is $\cos \theta = 1 - \alpha$, and for $\omega \approx \omega' \approx |\mathbf{k}|$,

$$E_{cm}^2 = |\mathbf{k}|^2 - |\mathbf{k}|^2 \cos \theta = |\mathbf{k}|^2 \alpha < \ell_{UV}^2, \tag{30}$$

and since $|\mathbf{k}| = \frac{\Lambda_0}{\alpha}$,

$$\sqrt{\alpha} > \Lambda_0 \ell_{\mathrm{UV}}. \tag{31}$$

This conclusion is the same as in the previous case eq. (27), where an explicit cutoff to the transverse momenta was imposed (22).

## 3 Examples

In this section we apply the SNEC bound eq.(3) to various situations of interest. First we examine the infinite geodesic length limit which reduces the bound to the ANEC integral. Then we examine long segments in the case of evaporating black holes, a situation where the NEC is violated. Finally we apply SNEC to the "long" wormhole of Maldacena, Milekhin and Popov. This is an interesting example as the complete geodesics are chronal thus achronal ANEC cannot be applied. However, it is possible to apply SNEC on shorter achronal segments.

### 3.1 The ANEC limit

Here we confirm that the SNEC bound of eq.(3) reduces to the ANEC at the limit of large support of the smearing function $g(\lambda)$. We introduce the rescaled function

$$g_{\lambda_0}(\lambda) = \frac{g(\lambda/\lambda_0)}{\sqrt{\lambda_0}}. \tag{32}$$

The rescaling is such that the normalization of $f$ does not depend on the choice of the parameter $\lambda_0$

$$\int_{-\infty}^{\infty} g^2(\lambda) d\lambda = \int_{-\infty}^{\infty} g_{\lambda_0}^2(\lambda) d\lambda = 1. \tag{33}$$

Then the bound of eq.(3) becomes

$$\int_{-\infty}^{+\infty} d\lambda g_{\lambda_0}^2(\lambda) \langle T_{kk} \rangle \geq -\frac{4B}{G_N} \int_{-\infty}^{+\infty} d\lambda \frac{1}{\lambda_0^3} \left( g'(\lambda/\lambda_0) \right)^2, \tag{34}$$

where we used

$$\frac{d}{d\lambda} g_{\lambda_0}(\lambda) = \frac{1}{\lambda_0^{3/2}} g'(\lambda/\lambda_0). \tag{35}$$

Multiplying both sides with $\lambda_0$ gives

$$\int_{-\infty}^{+\infty} d\lambda g^2(\lambda/\lambda_0) \langle T_{kk} \rangle \geq -\frac{4B}{G_N} \int_{-\infty}^{+\infty} d\lambda \frac{1}{\lambda_0^2} \left( g'(\lambda/\lambda_0) \right)^2. \tag{36}$$

Now we can take $\lambda_0 \to \infty$

$$\liminf_{\lambda_0 \to \infty} \int_{-\infty}^{+\infty} d\lambda g^2(\lambda/\lambda_0) \langle T_{kk} \rangle \geq 0. \tag{37}$$

The left hand side of eq.(37) is just the ANEC integral since $\sqrt{\lambda_0} g_{\lambda_0}(\lambda)$ converges uniformly to $g(0)$

$$\int_{-\infty}^{+\infty} d\lambda \langle T_{kk} \rangle \geq 0. \tag{38}$$

The rescaling method was first used with specific Lorentzian functions [11] and later for general ones to derive the ANEC, AWEC and ASEC for classical [37, 42] and quantum [37, 43] fields.

As an example we look at the Lorentzian function

$$g^2(\lambda) = \frac{1}{\pi} \frac{1}{\lambda^2 + 1}, \tag{39}$$

which is normalized in such a way that

$$\int_{-\infty}^{+\infty} g^2(\lambda)d\lambda = 1. \tag{40}$$

Now

$$g^2_{\lambda_0}(\lambda) = \frac{1}{\pi}\frac{\lambda_0}{\lambda^2 + \lambda_0^2}, \tag{41}$$

so that

$$\int_{-\infty}^{\infty} g^2_{\lambda_0}(\lambda)d\lambda = 1. \tag{42}$$

Using this function the bound of eq.(3) becomes

$$\int_{-\infty}^{+\infty} d\lambda \frac{\lambda_0^2}{\pi(\lambda^2 + \lambda_0^2)}\langle T_{kk}\rangle \geq -\frac{4B}{G_N}\int_{-\infty}^{+\infty} d\lambda \frac{1}{\pi}\frac{\lambda^2\lambda_0^2}{(\lambda_0^2 + \lambda^2)^2}. \tag{43}$$

Taking $\lambda_0 \to \infty$ gives the ANEC integral of eq.(38).

## 3.2 The evaporating black hole stress tensor

In previous work [4], we proposed that one should trust SNEC eq.(3) when smearing over null achronal segments of length much shorter than the radius of curvature. Here we provide some evidence supporting the generalization of our bound for longer regions of integration.

The case we examine is that of an evaporating black hole. In Schwarzschild geometry the radius of curvature is $r\sqrt{r/R_s}$ where $R_s$ is the Schwarzschild radius. Parametrizing the null geodesic with $r$ and picking a segment from $r$ to $2r$ we see that near the horizon the length of the segment is comparable to the radius of curvature [2].

We proceed to show that the stress tensor of an evaporating black hole satisfies SNEC, integrated over achronal, null geodesic segments. The relevant component of the stress tensor is [44]

$$T_{kk} \approx -\frac{N}{R_s^2 u^2}, \tag{44}$$

where $u$ is the usual outgoing null coordinate, and $R_s$ the Schwarszchild radius. More precise versions of eq.(44) can be found in [45], [46], and references therein. For our purposes, the stress tensor eq.(44) represents the negative energy density encountered by an observer traveling along a radial null geodesic outside of an evaporating black hole. We formulate our derivation as a theorem since stress tensors of this form are not specific to black holes, but they are in fact quite common.

**Theorem 3.1.** *For any $\mathbb{C}^1$ smearing function g, and a stress tensor of the form eq.(44), SNEC eq.(3) is satisfied.*

The stress tensor eq.(44) describes black holes that lie in the regime of semiclassical approximation, i.e. black holes for which $R_s^2 > \ell_{UV}^2 > NG_N$. This condition becomes relevant at the end of our proof.

*Proof.* Using the stress tensor of eq.(44) in eq.(3) we get

$$\frac{G_N N}{R_s^2}\int_{-\infty}^{\infty} du\frac{g^2}{u^2} \leq 4B\int_{-\infty}^{\infty} du\left(\frac{dg}{du}\right)^2. \tag{45}$$

---

[2]We thank Ken Olum for pointing out this example to us.

Substituting $e^y = u$, one obtains

$$\frac{G_N N}{R_s^2} \int_{-\infty}^{\infty} dy \; e^{-y} \; g^2 \leq 4B \int_{-\infty}^{\infty} dy \left(\frac{dg}{dy}\right)^2 e^{-y}. \tag{46}$$

Defining $\tilde{g} \equiv g e^{-y/2}$ we have

$$\frac{G_N N}{R_s^2} \int_{-\infty}^{\infty} dy \; \tilde{g}^2 \leq 4B \int_{-\infty}^{\infty} dy \left[\frac{1}{4}\tilde{g}^2 + \tilde{g}\frac{d\tilde{g}}{dy} + \left(\frac{d\tilde{g}}{dy}\right)^2\right]. \tag{47}$$

The second term on the r.h.s. of eq.(47) is a total derivative, which gives a vanishing boundary term. The third term on the r.h.s. of eq.(47) is manifestly positive. To conclude the proof, and assuming that $B$ is an order 1 number, we need to show that

$$\frac{G_N N}{R_s^2} \leq 1, \tag{48}$$

which is true for semiclassical black holes. $\qquad\square$

## 3.3 The Maldacena-Milekhin-Popov wormhole

In their recent work [6] Maldacena, Milekhin and Popov presented a wormhole solution in four dimensions. It is what we call a "long" wormhole, meaning that it takes longer for an observer to go between two points in spacetime if they travel inside the wormhole than if they travel in ambient space. Thus this wormhole does not lead to causality violations. The wormhole is a solution of an Einstein–Maxwell theory with charged massless fermions which give rise to negative energy, necessary for the existence of a wormhole. Here we examine the wormhole in relation to the SNEC bound. In particular we are interested in whether or not the wormhole solution saturates the proposed bound in four and two dimensions.

In [6] the wormhole interior is given as a first approximation, from matching the $AdS_2 \times S_2$ geometry

$$ds^2 = r_e^2 \left[-(\rho^2 + 1)d\tau^2 + \frac{d\rho^2}{\rho^2 + 1} + d\Omega_2^2\right], \tag{49}$$

to the charged black hole geometry. In eq.(49), $r_e$ parametrizes the size of the "mouth" of the wormhole. The matching conditions are

$$\tau = \frac{t}{\ell}, \quad \rho = \frac{\ell(r - r_e)}{r_e^2}, \quad \text{with} \quad 1 \ll \rho, \quad \frac{r - r_e}{r_e} \ll 1, \quad 1 \ll \frac{\ell}{r_e}, \tag{50}$$

where $\ell$ is a free length scale with is later identified as the length of the wormhole.

### 3.3.1 Four-dimensional bound

The full length of the geodesic inside the wormhole is chronal, since $\pi\ell > d$, where $d$ is the approximate distance of the mouths in ambient space. But how long is the maximum achronal segment measured in the dimensionless coordinate $\rho$? The condition is that the length of the achronal segment has to be shorter than the rest of the geodesic inside the wormhole added to the distance between the mouths in ambient space so

$$\Delta\rho\big|_{WH} < \Delta\rho\big|_{OUT}. \tag{51}$$

Let's assume that the segment's middle is at $\rho = 0$ and it stretches form $-\rho_0/2$ to $\rho_0/2$. Then

$$\Delta\rho\big|_{WH} = \int_{-\rho_0/2}^{\rho_0/2} \frac{d\rho}{1 + \rho^2} = 2\arctan(\rho_0/2). \tag{52}$$

To find $\Delta\rho\big|_{OUT}$ we have to add the remaining pieces inside the wormhole and the distance in ambient space which in the $\rho$ coordinate is approximately [3] $d/\ell$. This is because of the matching in eq.(5.19) in [6]. We have

$$\Delta\rho\big|_{OUT} = \int_{-\infty}^{-\rho_0/2} \frac{d\rho}{1+\rho^2} + \frac{d}{\ell} + \int_{\rho_0/2}^{\infty} \frac{d\rho}{1+\rho^2} = \pi - 2\arctan(\rho_0/2) + \frac{d}{\ell}. \tag{53}$$

From condition eq.(51) we have

$$\rho_0 < 2\tan\left(\frac{\pi}{4} + \frac{d}{4\ell}\right). \tag{54}$$

Inserting the maximum value of $d = \pi\ell/2.35$, extracted numerically from the relationship between $\pi\ell$ and d in [6], we have that $\rho_0 < 4.13$. If we could approach the limit of a "short" wormhole $d \to \pi\ell$ then $\rho_0$ is allowed to become larger.

The quantum contribution to the energy density in the throat was calculated in [6] and it has the general form

$$T_{tt} = -\frac{q}{24\pi^2 r_e^2}\left(-\frac{1}{4\ell^2} + \int_0^2 \frac{dv}{2}\frac{\pi^2}{(\pi\ell + df(v))^2}\right), \tag{55}$$

where the first term is a contribution from the conformal anomaly, and the second is the Casimir energy. The parameter $q$ is the number of two-dimensional fermionic fields. A key idea here is that one four dimensional field gives rise to a $q \gg 1$. The function $f(v)$ is the length of the field lines and $v$ is a ratio of angular momentum quantum numbers of the fermions. For $d = \pi\ell/2.35$ we have

$$T_{tt} = T_{xx} = -\frac{q}{12\pi^2 r_e^2 \ell^2}\left[-\frac{1}{4} + \int_0^2 \frac{dv}{2}\left(1 + \frac{f(v)}{2.35}\right)^{-2}\right] \equiv -\frac{q}{12\pi^2 r_e^2 \ell^2}A, \tag{56}$$

where $A$ is a constant of order 1.

If we introduce the following coordinates $\sigma$ and $\lambda$

$$d\sigma \equiv d\tau = \frac{d\rho}{(1+\rho^2)}, \qquad d\lambda \equiv (1+\rho^2)d\tau = d\rho, \tag{57}$$

the form of the SNEC bound eq.(3) becomes

$$\int_{\infty}^{\infty} g^2(\lambda)\left(\frac{dx^-}{d\lambda}\right)^2 T_{--} \geq -\frac{4B}{G_N}\int_{\infty}^{\infty} d\lambda\, g'(\lambda)^2, \tag{58}$$

where $x_- = \tau - \sigma$. Taking into account that $\tau = t/\ell$, one can check that $T_{--} = 2\ell^2 T_{tt}$. Then eq.(58) becomes

$$\int_0^{\infty} d\rho\, g(\rho)^2 \frac{1}{(1+\rho^2)^2} T_{--} \geq -\frac{4B}{G_N}\int_0^{\infty} d\rho\, g'(\rho)^2. \tag{59}$$

The stress-energy tensor is a constant so we can define

$$C \equiv \frac{|T_{--}|G_N}{4B} = \frac{g^2}{24B\pi^3 q}A, \tag{60}$$

---

[3]Here we calculate the distance between the black hole horizons. However $r_e \ll d$ so this is a valid approximation for the distance in ambient space.

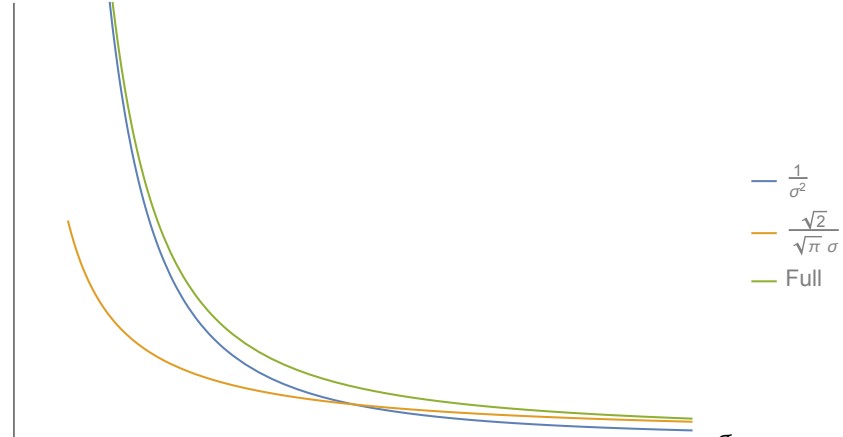

Figure 2: Qualitative plot of the right hand side of eq.(64) and asymptotic behavior for large and small $\sigma$.

since

$$r_e^2 \equiv \frac{\pi q^2 \ell_{\mathrm{pl}}^2}{g^2}, \qquad \text{and} \qquad \ell_{\mathrm{pl}} \equiv \sqrt{G_N}, \tag{61}$$

where $g$ is the gauge coupling [6]. Considering that $g$ and $A$ are order 1 numbers, $C \sim 1/Bq$.

Using eq.(60), we rearrange eq.(59)

$$C \int_0^\infty d\rho\, g(\rho)^2 \frac{1}{(1+\rho^2)^2} \leq \int_0^\infty d\rho\, g'(\rho)^2, \tag{62}$$

which for a Gaussian function of width $\sigma$ becomes

$$C \int_0^\infty d\rho\, e^{-\frac{2\rho^2}{\sigma^2}} \frac{1}{(1+\rho^2)^2} \leq \frac{1}{\sigma^4} \int_0^\infty d\rho\, 4\rho^2\, e^{-\frac{2\rho^2}{\sigma^2}}. \tag{63}$$

A direct computation of the integrals gives

$$C \leq \sqrt{2} \left( 2 + \sigma^{-1} e^{2/\sigma^2} \sqrt{\pi} (\sigma^2 - 4) \operatorname{erf}[\sqrt{2}/\sigma] \right)^{-1}, \tag{64}$$

where erf is the error function. To investigate the behavior of the right hand side of eq.(64) we examine small and large values of $\sigma$.

For $\sigma \ll 1$ we have

$$C \leq \frac{1}{\sigma^2} + \mathcal{O}(\sigma^4), \tag{65}$$

while for $\sigma \gg 1$ we have

$$C \leq \frac{\sqrt{2}}{\sqrt{\pi}\sigma} + \mathcal{O}\left(\frac{1}{\sigma^3}\right). \tag{66}$$

A plot of the r.h.s. of eq.(64) is given in Fig. 2.

The width of the Gaussian $\sigma$, is essentially the smearing length, i.e. the length of the achronal segment on which one observes the negative energy density. The length of this segment for the wormhole is bounded by $\sigma \equiv \rho_0 < 4.13$. Since $q$ is the number of species, and one needs $q \gg 1$ to be in the semiclassical limit (to not worry about quantum corrections that would destabilize the solution), SNEC is respected. That is true however, provided that $B$ is an order one number; if $B$ is instead of order $1/q$ the bound may indeed be *saturated*.

### 3.3.2 Two-dimensional bound

Although the wormhole of Ref. [6], may not saturate SNEC for most $q$, we prove here an $(1+1)$-$d$ bound that does. To be able to study the wormhole using a $(1+1)$-$d$ bound, we focus on the $AdS_2$ part of the geometry. Fewster and Hollands [8] have proven the following null smeared QEI for certain interacting CFT's in $(1+1)$ Minkowski spacetime:

$$\int_{-\infty}^{+\infty} f(\rho)\langle\hat{T}_{ab}(\rho))\rangle_\omega k^a k^b d\rho \geq -\frac{c}{48\pi}\int_{-\infty}^{+\infty}\frac{(f')^2}{f}d\rho\,, \tag{67}$$

where $c$ is the total central charge of the theory. Blanco at al. [47] confirmed and slightly improved that bound using modular Hamiltonians. Surprisingly, this bound has yet to be generalized to curved space. Following work of Flanagan [28], we provide a simple proof of eq.(67) for spacetimes *globally conformal to Minkowski*, and *arbitrary* smearing functions. The detailed proof is given in Appendix A. We should note that while $AdS_2$ is not globally conformal to Minkowski the bound can be used approximately in the wormhole case as long as we stay away from the boundaries. So have

$$\int_{-\infty}^{\infty}d\rho\, g(\rho)^2\frac{1}{(1+\rho^2)^2}T_{--}^{2d}\leq\frac{c}{12\pi}\int_{-\infty}^{\infty}d\rho\, g'(\rho)^2\,. \tag{68}$$

The $(1+1)$-d stress tensor is given by

$$T_{--}^{2d}=r_e^2\, T_{--}\propto q\,, \tag{69}$$

where $r_e$ is the size of the $S^2$ in the near horizon $AdS_2\otimes S^2$ geometry. Equation eq.(68) becomes

$$\frac{q}{c}\int_0^{\infty}d\rho\, g(\rho)^2\frac{1}{(1+\rho^2)^2}\leq\int_0^{\infty}d\rho\, g'(\rho)^2\,. \tag{70}$$

As we have mentioned $q$ is approximately proportional to the number of species of fields. In the CFTs of interest, this role is played by $c$, so $q/c$ is of order one. As in 3.3.1 we calculate the integrals in eq.(70) for a Gaussian of width $\sigma$ to obtain

$$\frac{q}{c}\leq\sqrt{2}\left(2+\sigma^{-1}e^{2/\sigma^2}\sqrt{\pi}(\sigma^2-4)\,\mathrm{erf}[\sqrt{2}/\sigma]\right)^{-1}\,. \tag{71}$$

For $\sigma\ll 1$ we have

$$\frac{q}{c}\leq\frac{1}{\sqrt{2}\sigma^2}+\mathcal{O}(\sigma^4)\,, \tag{72}$$

while for $\sigma\gg 1$ we have

$$\frac{q}{c}\leq\frac{\sqrt{2}}{\sqrt{\pi}\sigma}+\mathcal{O}\left(\frac{1}{\sigma^3}\right)\,. \tag{73}$$

Since $q/c$ is an order one number, and $\sigma<4.13$ since for a Gaussian it is equal to the length of the achronal segment, this $(1+1)$-d bound can be easily *saturated*.

## 4 A Penrose-type singularity theorem

An important application of the SNEC bound is the proof of a singularity theorem. The question of "whether the semi-classical effects of negative energy invalidate the singularity theorems, before quantum gravity effects become significant" was suspected to have a negative answer [48]. Here we present a proof of a null singularity theorem using a condition obeyed by quantum fields.

Ref. [36] proved a singularity theorem for null geodesic incompleteness with the NEC replaced by a weaker condition inspired by QEIs using index form methods. After we state that condition we show that the SNEC bound is a bound of the same form. Then we specify the required initial condition that leads to geodesic incompleteness. Finally we examine a particular example, that of the spherically symmetric evaporating black hole.

Let $P$ be a future converging achronal spacelike submanifold of $M$ of co-dimension 2 with mean normal curvature vector field $H^\mu = H\hat{H}^\mu$ where $\hat{H}^\mu$ is a future-pointing timelike unit vector. Then let $\gamma$ be a future-directed null geodesic emanating normally from $P$. As in all null geodesics we need to specify an affine parametrization for $\gamma$.

We extend $\hat{H}_\mu$ by parallel transporting along $\gamma$. Next we choose an affine parameter $\lambda$ on $\gamma$, such that $\hat{H}_\mu d\gamma^\mu/d\lambda = 1$. Then define $\ell$, the length of the geodesic with respect to $\lambda$. Now we can state the condition required in [36]

$$\int_0^\ell g(\lambda)^2 R_{kk} d\lambda \geq -Q_m(\gamma)\|g^{(m)}\|^2 - Q_0(\gamma)\|g\|^2, \tag{74}$$

where $Q_m$ and $Q_0$ are constants that depend on the geodesic $\gamma$ and $m$ a positive integer. The notation $\|\cdot\|$ denotes the $L^2$ norm so

$$\|g\|^2 \equiv \int_{-\infty}^\infty d\lambda\, g(\lambda)^2. \tag{75}$$

The inequality of eq. (3) is a bound on the expectation value of components of the stress-energy tensor. But singularity theorems require a geometric assumption which, in the case of Penrose-type theorems, is a bound on $R_{kk}$. Classically the Einstein equation connects curvature to the stress-energy tensor. Semiclassically, the semiclassical Einstein equation (SEE) equates the expectation value of the stress-energy tensor with the classical Einstein tensor

$$8\pi G_N \langle T_{\mu\nu}\rangle_\omega = G_{\mu\nu}. \tag{76}$$

With the use of SEE we assume that we have a self-consistent solution, which includes a state $\omega$ and a metric $g_{\mu\nu}$ that satisfy eq.(76). Using the SEE the bound of eq.(3) can be written as

$$\int_{-\infty}^\infty g(\lambda)^2 R_{kk} d\lambda \geq -32\pi B\|g'(\lambda)\|^2. \tag{77}$$

Then this is a bound of the form of eq.(74) with $m = 1$, $Q_1 = 32\pi B$ and $Q_0 = 0$.

## 4.1 Mean normal curvature

Ref. [36] has two scenarios to describe all possible initial conditions: in scenario 1, initially the NEC is satisfied for an affine length $\ell_0$, short compared to the one for the formation of a focal point $\ell$. In scenario 2 this requirement is dropped and instead conditions are imposed on the null contracted Ricci tensor for small negative values of the affine parameter. For scenario 2, perhaps counterintuitively, negative null energy in this region leads to smaller required initial contraction because this negative energy must be over-compensated by positive energy, an effect known as "quantum interest" [49].

The goal is to find the mean normal curvature $H$ of $P$ required to have null geodesic incompleteness for the bound of eq.(77).

For scenario 1 we suppose that initially the NEC is satisfied, and so let $\rho = R_{kk}$ be an initially positive function, $\rho \geq \rho_0 \geq 0$ on $[0, \ell_0]$ for some $0 < \ell_0 < \ell$. Then we can use Lemma 4.1 of Ref. [36] with $m = 1$, $Q_0 = 0$, $A_1 = 1/3$, $B_1 = C_1 = 1$, which gives

**Lemma 4.1.** *For $\rho$ satisfying eq.(77) on $[0,\ell]$ we have that if*

$$-2H|_{\gamma(0)} \geq \nu^* \equiv -\frac{2}{3}\rho_0\ell_0 + \frac{Q_1}{\ell_0} + \frac{2+Q_1}{\ell-\ell_0},\tag{78}$$

*then $\gamma$ contains a focal point before $\ell$.*

Assuming that $\ell \gg \ell_0$ we can discard the last term and get

$$\nu^* \leq -\frac{2}{3}\rho_0\ell_0 + \frac{Q_1}{\ell_0}.\tag{79}$$

Then for given $\ell_0$ and $\rho_0$ we can calculate the required initial contraction to have a focal point before $\ell$. As expected for larger $\ell_0$ and $\rho_0$ smaller initial contraction is required. However, for $\nu^* < 3/\ell_0$ the original Penrose theorem should be used instead.

Turning to scenario 2 we drop the assumption that the NEC holds. We instead extend $\gamma$ to $\gamma : [-\ell_0,\ell] \to M$ and assume that eq.(77) holds on the extended geodesic. Then we define $\rho_{\max} = \max_{[-\ell_0,0]}\rho$ and we can use Lemma 4.7 of Ref. [36] with $m = 1$, $Q_0 = 0$, $A_1 = 1/3$, $B_1 = C_1 = 1$, which gives

**Lemma 4.2.** *For $\rho$ satisfying eq.(77) on $[-\ell_0,\ell]$ if*

$$-2H \geq L_1(\ell) + L_2(\ell_0),\tag{80}$$

*then there is a focal point to P along $\gamma$ in $[0,\ell]$. Here*

$$\hat{L}_1(\ell') = \frac{Q_1+2}{\ell'},\tag{81}$$

$$\hat{L}_2(\ell_0') = \frac{Q_1}{\ell_0'} + \frac{1}{3}\rho_{\max}\ell_0',\tag{82}$$

*and*

$$L_1(\ell) = \min_{\ell'\in(0,\ell]}\hat{L}_1(\ell'), \qquad L_2(\ell_0) = \min_{\ell_0'\in(0,\ell_0]}\hat{L}_2(\ell_0').\tag{83}$$

It is easy to see that $\hat{L}_1$ is minimized for $\ell' = \ell$. For $\hat{L}_1$ and $\rho_{\max} < 0$ it is minimized for $\ell_0' = \ell_0$ while for $\rho_{\max} > 0$ for $\ell_0' = \sqrt{\frac{3Q_1}{\rho_{\max}}}$, if smaller than $\ell_0$. In the case that $\ell_0' = \ell_0$ gives the minimum we have

$$-2H \geq \frac{Q_1+2}{\ell} + \frac{Q_1}{\ell_0} + \frac{1}{3}\rho_{\max}\ell_0.\tag{84}$$

## 4.2 Application to evaporating black holes

In his seminal work [3], Penrose proved the first singularity theorem which applies to a classical black hole spacetime. In an evaporating black hole spacetime, where the NEC is violated, the original Penrose theorem cannot be used. This provides us with the perfect opportunity to test the previous two Lemmas.

We assume that the metric is well-approximated by Schwarzschild geometry near the horizon. The metric inside the horizon is

$$ds^2 = \left(\frac{R_s}{r}-1\right)dt^2 - \left(\frac{R_s}{r}-1\right)^{-1}dr^2 + r^2d\Omega^2,\tag{85}$$

where $R_s$ is the Schwarzschild radius. As described previously, to fix the affine parametrization of the null geodesic we use the mean normal curvature vector field of the hypersurface $P$.

We focus on spherically symmetric hypersurfaces $P$, so that the hypersurface is defined by Schwarzschild coordinates $(t_p, r_p)$. Because the metric is static and spherically symmetric, the mean normal curvature vector field is purely in the $r$ direction. Requiring $\hat{H}_\mu d\gamma^\mu/d\lambda = 1$ we have

$$\lambda = \frac{r - r_P}{\sqrt{\left(\frac{R_s}{r_P} - 1\right)}}, \tag{86}$$

where $r_P$ is the radial coordinate of the hypersurface $P$.

The mean normal curvature $H$ of hypersurfaces with constant $r$ and $t$ changes as we go further inside the black hole. Inside the horizon, the mean normal curvature of our surfaces $P$ is given by the function [50]

$$H(r_P) = -\frac{1}{r_P}\sqrt{\frac{R_s}{r_P} - 1}\,. \tag{87}$$

We start with scenario 2 and eq.(84). First we assume $\rho_{\max} < 0$ so that the NEC is violated everywhere in $\ell_0$ and so we can drop the last term, obtaining

$$H < -\frac{Q_1}{2\ell} - \frac{1}{\ell} - \frac{Q_1}{2\ell_0}\,. \tag{88}$$

The two parameters are the maximum affine parameter for the formation of the singularity $\ell$ and the length of the affine parameter that the NEC is violated $\ell_0$.

We need to pick the point where we start having NEC violation. To include $\ell_0$ all null geodesics emanating normally from $P$ need to be able to extend to that point. While ingoing radial null geodesics can be extended backwards to large $r$, outgoing geodesics can only be extended back as far as the past horizon, where the Unruh state is singular. Physically, this region of the spacetime should be replaced by whatever matter collapsed to form the black hole.

Since the congruence of outgoing geodesics cannot be extended further than the event horizon, we define

$$R_s - r_P \equiv x R_s\,, \qquad 0 < x < 1\,. \tag{89}$$

The setup is shown in Fig. 3. The $\ell$ can have as a minimum $(1 - x)R_s$, meaning that the singularity is located exactly at $r = 0$; we can also consider sending $\ell \to \infty$, meaning we have no information about the location of the singularity.

We define $y$ by demanding that the affine distance $\ell$ is a coordinate distance $y R_s$, so we have

$$\ell_0 = x R_s \left(\frac{1}{1-x} - 1\right)^{-1/2} \quad \text{and} \quad \ell = y R_s \left(\frac{1}{1-x} - 1\right)^{-1/2}\,. \tag{90}$$

Now we can pick a point $(1 - x)R_s$ from $r = 0$ where the mean normal curvature is smaller than the one on the right hand side of eq.(88). Equating the two expressions of eq.(88) and eq.(87) for $H$ we have

$$\frac{1}{1-x} = \frac{Q_1}{2y} + \frac{1}{y} + \frac{Q_1}{2x}\,. \tag{91}$$

The two extreme cases are for $y = 1 - x$ where we have no solutions and $y \to \infty$ where $x = Q_1/(2 + Q_1)$. A plot of $x$ for different values of $y$ is shown in Fig. 4 for two different values of $Q_1$.

The Ref. [5] value of $B = 1/32\pi$ translates to $Q_1 = 1$. Using this value for $Q_1$, we find that the minimum $x$ is $1/3$. Therefore, we can prove the singularity theorem for any surface $P$ with

$$r_P \le \frac{2}{3}R_s\,. \tag{92}$$

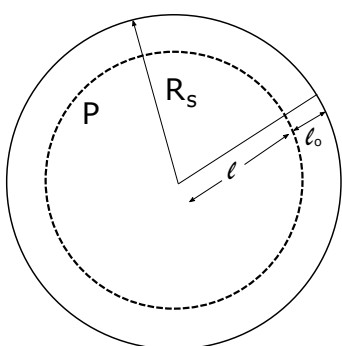

Figure 3: Schematic representation of a Schwarzschild black hole and the parameters in scenario 2. The dashed circle is constant $r$ and $t$ hypersurface $P$. Distance $\ell_0$ is from the point where the NEC starts being violated, and distance $\ell$ is from $P$ to the singularity (pictured here at $r = 0$).

As we discussed in Sec. 2.2 there is also strong motivation to use a value of $B \ll 1$ and so $Q_1 \ll 1$. For small $Q_1$, we have a singularity theorem for spheres $P$ with

$$R_s - r_P \gtrapprox R_s \frac{Q_1}{2}, \qquad \text{for } Q_1 \ll 1. \tag{93}$$

The requirement that the NEC is violated for an affine parameter length comparable to the Schwarzschild radius[4] is a strong requirement. However, it is not necessary. First we remember that scenario 2 works for both negative and positive $\rho_{\max}$. Analytic approximations of the null energy near a Schwarzschild black hole horizon [46] show that the maximum values (positive or negative) do not exceed $\sim 10^2 G_N / R_s^4$ for a single field. For $N$ fields we have $\sim 10^2 N G_N / R_s^4$. To be in the semiclassical regime we assume $N G_N < \ell_{uv}^2$. Astrophysical black holes have $R_s \gg \ell_{uv}$. Then $\sqrt{3 Q_1 / \rho_{\max}} \sim R_s^2 / \sqrt{N G_N} > R_s^2 / \ell_{uv} \gg R_s > \ell_0$, so eq.(84) holds. Its last term is of order $N G_N / R_s^3 < \ell_{uv} / R_s^3 \ll 1/R_s$. So even though in the case of $\rho_{\max} > 0$ we need a larger mean normal curvature on $P$, that contribution is orders of magnitude smaller than the other terms. Then the previous discussion can be extended to cases where the NEC in not violated everywhere in $\ell_0$ with very small corrections.

Turning to scenario 1 (Lemma 4.1) we first we note that since $\rho_0 > 0$ we can discard the first term of eq.(78) so we have

$$H \leq -\frac{Q_1}{2\ell_0} - \frac{2 + Q_1}{2(\ell - \ell_0)}. \tag{94}$$

Here $\ell_0$ is the distance for which the NEC is obeyed and it is unknown so we set it to coordinate distance $x R_s$ from $P$. The distance from $P$ to $r = 0$ is also unknown here so we set it equal to $z R_s$ (see Fig. 5). The distance $\ell$ for singularity formation varies from $z R_s$ to $\infty$ so we use the variable $y$ as before. Then $\ell_0 = x R_s \left( \frac{1}{z} - 1 \right)^{-1/2}$ and $\ell = y R_s \left( \frac{1}{z} - 1 \right)^{-1/2}$.

To apply Lemma 4.1 we need the $H$ of eq.(94) to be smaller in absolute value than the one in eq.(87) for $r = y R_s$. Then eq.(94) becomes

$$\frac{1}{z} \geq \frac{Q_1}{2x} + \frac{2 + Q_1}{2(y - x)}. \tag{95}$$

There are three variables to vary: $z$ (distance of $P$ from $r = 0$), $x$ (region of NEC obeyed) and $y$ (distance to singularity). Not all values of them are possible though. First we note that

---

[4]For $Q_1 = 1$ the minimum violating affine length is $R_s/3$.

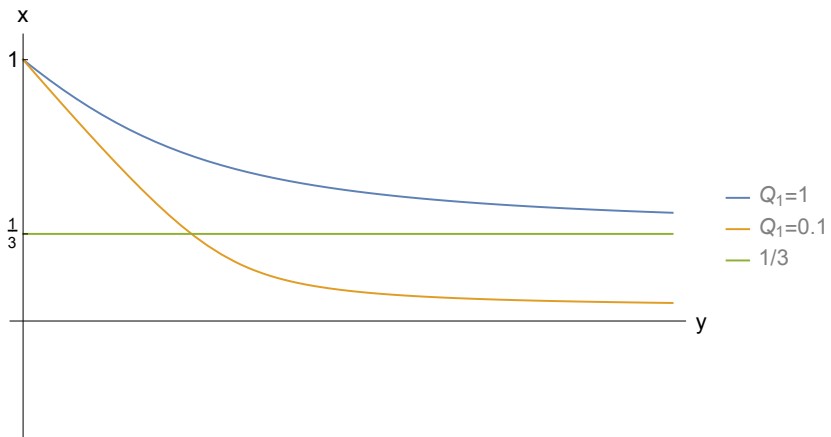

Figure 4: Required value of $x$ to have a singularity for different values $y$. For $Q_1 = 0.1$ the minimum value is much smaller compared to $Q_1 = 1$.

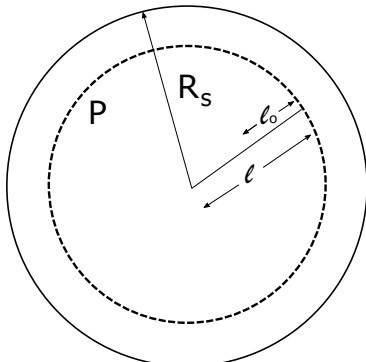

Figure 5: Schematic representation of a Schwarzschild black hole and the parameters in scenario 1. The dashed circle is constant $r$ and $t$ hypersurface $P$. Distance $\ell_0$ is measured from the hypersurface $P$ until the NEC starts being violated. Distance $\ell$ is from $P$ to the singularity (pictured here at $r = 0$).

there is no solution for $y \leq z$ (singularity at most at $r = 0$). However, there are solutions for larger values of $y$. Second the NEC should not be obeyed everwhere from $P$ to $r = 0$ (then the original Penrose theorem applies). Third the location of $P$ is from $r = 0$ to $r = R_s$. Mathematically these conditions are

$$y \in [z, \infty), \qquad x < z, \quad \text{and} \quad z \in [0, 1]. \tag{96}$$

At different physical situations one might want to minimize $x$ (small range of NEC obeyed) or maximize $z$ ($P$ closer to the horizon). As expected for smaller values of $Q_1$ it is easier to have both small $x$ and large $z$. Figure 6 shows all cases for $Q_1 = 1$ and $Q_1 = 0.1$.

Looking at both scenarios we can see that there are several physically interesting situations where the Lemmas presented can be applied. Specifically we can have a singularity when the NEC is violated for part of the black hole spacetime, a case where the Penrose theorem doesn't apply. Of course one could consider a situation where the NEC is violated close to the horizon and then obeyed with very large positive energies for only a short affine parameter where our Lemmas do not apply. We should note though that does not mean necessarily singularity avoidance. These cases could be minimized with a stronger SNEC bound and in particular with a small $B$.

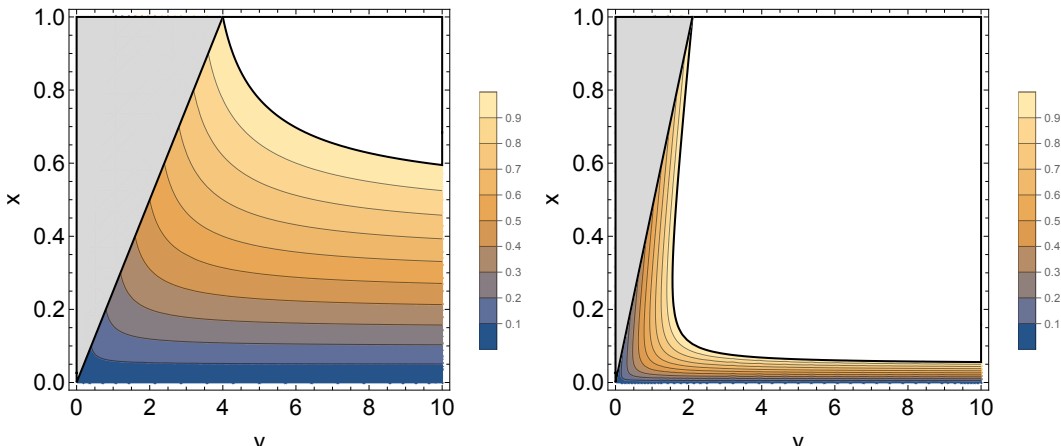

Figure 6: Density plot of the distance $z$ from $r = 0$ to $P$ in terms of $x$ (NEC obeyed) and $y$. The gray area represents a region where the $z > x$ condition is not met while the white area the region where the $z < 1$ condition is not met. $y > z$ everywhere in this region. The left plot is for $Q_1 = 1$ while the right for $Q_1 = 0.1$.

### 4.3 Comparison to quantum singularity theorem

An alternative singularity theorem that allows NEC violations was proposed by Wall [51]. It assumes the the generalized second law (GSL) [52] and the existence of a quantum trapped surface, defined in terms of the generalized entropy [53].

It is not straightforward to compare the assumptions of the two theorems as the GSL is a very different condition from the SNEC. What we can do is compare the location of the quantum trapped surface with the location of the surface with the required mean normal curvature.

Penington [54] calculated the location of the quantum extremal surface of a spherically symmetric evaporating black hole. The location of the surface was found to be

$$r = R_s - \frac{\beta}{\pi}\left|\frac{\partial R_s}{\partial v}\right|, \tag{97}$$

where $v$ is in Eddington-Finkelstein coordinates and $\beta$ is the inverse temperature of the black hole. The location of the horizon in respect to the apparent horizon ($R_s$) is

$$r_{\text{hor}} = R_s - \frac{\beta}{2\pi}\left|\frac{\partial R_s}{\partial v}\right|. \tag{98}$$

The evaporation rate in four dimensions is

$$\left|\frac{\partial R_s}{\partial v}\right| = \frac{c_{\text{evap}}G_N}{24\pi\beta R_s}, \tag{99}$$

where $c_{\text{evap}} = N_b + N_f/2 \equiv N$ is the number of species. Then

$$r_Q = r_{\text{hor}} - \frac{N G_N}{48\pi^2 R_s}. \tag{100}$$

We can estimate the distance of the quantum extremal surface from the event horizon for semiclassical black holes using $N G_N \leq \ell_{uv}^2$. Then

$$r_Q = r_{\text{hor}} - R_s \epsilon, \qquad \text{with} \quad \epsilon \lesssim (\ell_{uv}/R_s)^2. \tag{101}$$

For astrophysical black holes $\epsilon$ is expected to related to the black hole entropy via

$$\epsilon \sim \frac{1}{S}. \tag{102}$$

We should note that this calculation is only valid for slowly evaporating black holes.

This is a much shorter distance from the classical horizon (the proper distance lies between $\ell_{UV}$ and the Planck scale) compared to our estimates for $Q_1 = 1$.

It remains an open question *why* the quantum singularity theorem delivers a much stronger singularity theorem than SNEC. One possibility is that the quantum focusing theorem simply requires stronger assumptions; for example, the proof relies on the generalized second law (GSL), which is violated during the later stages of Hawking radiation. Another interesting possibility is that SNEC may be a sub-optimal bound, and that a stronger bound could be proven.

One benefit of our approach is that the singularity theorem proposed here gives an estimate for the location of the singularity where no information is given in the case of the quantum singularity theorem. A further comparison of the conditions of the two theorems and specifically the requirement of the GSL would be valuable.

# 5 Conclusion

In this work we have given both motivation and useful applications for the smeared null energy condition (SNEC) introduced in [4]. We showed that a field theory version of SNEC can be derived in Minkowski spacetime for free and super-renormalizable theories. This version of the bound addresses the counterexample of Fewster and Roman [7]. We showed that SNEC reduces to the averaged null energy condition (ANEC) at the limit of large support, and that it is a valid bound over segments comparable to the radius of curvature for evaporating black holes. After detailed analysis, we concluded that SNEC is not saturated in the case of the Maldacena-Milekhin-Popov wormhole in four dimensions. However, we showed that this wormhole *does* saturate the analogous dimensionally reduced 2d QEI whose curved space version we proved here. Finally, we proved a Penrose-type singularity theorem using SNEC and applied this theorem to establish that spacetimes that approximate the Schwarzschild solution near the horizon must contain a singularity.

Even though there is strong motivation for it, SNEC remains unproven in the case of general curved spacetimes and interacting fields. An important open question is whether or not there are correction terms to the SNEC bound dependent on the curvature. In the case of quantum energy inequalities over timelike curves such terms appear [17], but they do not in the null integrated bounds in two dimensions [28]. The interacting case is more challenging as QEI results are few, but perhaps methods such as those used for proving ANEC [20] could be useful here.

In general, SNEC can be used in scenarios when achronal ANEC cannot be used, since it can be applied to smaller achronal pieces of chronal curves. Examples include different traversable wormholes such as the one proposed by Gao, Jafferis and Wall [55]. A different direction is the case of the area theorem and its tension with black hole evaporation. Comparison of SNEC with different proposed bounds such as the quantum null energy condition [24] or the GSL [56] are an interesting direction. In a different application, bounce cosmology scenarios often require the violation of singularity theorems [57, 58] which is attributed to the violation of NEC by quantum fields. Using a singularity theorem obeyed by quantum fields, these models can be critically re-examined.

We do not have an example where SNEC is saturated. Therefore, it may be that a stronger bound than SNEC is true. It would be interesting to find an example saturating SNEC, or

to suggest a stronger bound. One example where a stronger bound was proven was in [21] where they proved a positivity result for incomplete (but maximally extended) achronal null geodesics in $AdS^2 \times Sd_2$.

Finally, it would be very nice to derive a version of the field theory SNEC that is independent of the UV cutoff. This will require smearing over additional directions. Some results along these lines appear in [39].

## Acknowledgements

The authors would like to thank Srivatsan Balakrishnan, Thomas Faulkner, Chris Fewster, Eanna Flanagan, Jackson Fliss, Diego Hofman, Manthos Karydas, Juan Maldacena, Ken Olum and Alex Vilenkin for useful discussions.

**Funding information**  BF and E-AK are supported by the ERC Consolidator Grant QUANTIV-IOL. This work is part of the $\Delta$ ITP consortium, a program of the NWO that is funded by the Dutch Ministry of Education, Culture and Science (OCW).

## A  Proof of two-dimensional curved spacetime null QEI for interacting CFTs

Following the derivation of Flanagan [28], we generalize the result of Fewster-Hollands [8] to spacetimes *globally conformal to Minkowski*. That is, we prove the result of [8]

$$\int_{-\infty}^{+\infty} f(\lambda)\langle \hat{T}_{ab}(\lambda)\rangle_\omega k^a k^b d\lambda \geq -\frac{c}{48\pi}\int_{-\infty}^{+\infty}\frac{(f')^2}{f}d\lambda, \tag{103}$$

which holds for a class of interacting quantum fields, namely the unitary, positive energy conformal field theories with stress-energy tensor, and in a large class of curved backgrounds, *for arbitrary smearing functions*.

First we have to show that the integral

$$J_\gamma[g_{ab}, k^a, f, \omega] = \int_{-\infty}^{+\infty} f(\lambda)\langle \hat{T}_{ab}(\lambda)\rangle_\omega k^a k^b d\lambda + \frac{c}{48\pi}\int_{-\infty}^{+\infty}\left(\frac{df(\lambda)}{d\lambda}\right)^2 \frac{1}{f}d\lambda, \tag{104}$$

is invariant

$$J_\gamma[\bar{g}_{ab}, \bar{k}^a, \bar{f}, \bar{\omega}] = J_\gamma[g_{ab}, k^a, f, \omega], \tag{105}$$

under the following conformal trasnformation

$$\bar{g}_{ab} = e^{2\sigma} g_{ab}, \tag{106}$$

$$\bar{k}^a = e^{-2\sigma} k^a, \tag{107}$$

$$\bar{f} = e^{2\sigma} f, \tag{108}$$

$$d\bar{\lambda} = e^{2\sigma} d\lambda, \tag{109}$$

where $k^a = \left(\frac{\partial}{\partial_\xi}\right)^a$. The stress-tensor transforms as follows [59]

$$\langle \hat{T}_{ab}\rangle_{\bar{\omega}} = \langle \hat{T}_{ab}\rangle_\omega + \frac{c}{12\pi}\left[\nabla_a\nabla_b\sigma - \nabla_a\sigma\nabla_b\sigma - g_{ab}\nabla_c\nabla^c\sigma + \frac{1}{2}g_{ab}(\nabla\sigma)^2\right]. \tag{110}$$

Since we are looking at the null contracted version of eq.(110), the last two terms vanish.

*Proof.*

$$
\boxed{J_\gamma[\bar{g}_{ab},\bar{k}^a,\bar{f},\bar{\omega}]} = \int_{-\infty}^{+\infty} \bar{f}\,\langle\hat{T}_{ab}\rangle_{\bar{\omega}}\,\bar{k}^a\bar{k}^b\,d\bar{\lambda} + \frac{c}{48\pi}\int_{-\infty}^{+\infty} \frac{\left(d\bar{f}/d\bar{\lambda}\right)^2}{\bar{f}}\,d\bar{\lambda}
$$

$$
= \int_{-\infty}^{+\infty} f\,\langle\hat{T}_{ab}\rangle_{\omega}\,k^a k^b\,d\lambda + \frac{c}{48\pi}\int_{-\infty}^{+\infty}\left(\frac{d(fe^{-2\sigma})\,e^{-2\sigma}}{d\lambda}\right)^2\frac{1}{f}\,d\lambda
$$

$$
+ \frac{c}{12\pi}\int_{-\infty}^{+\infty} f k^a k^b\left[\nabla_a\nabla_b\sigma - \nabla_a\sigma\nabla_b\sigma\right]d\lambda
$$

$$
= \int_{-\infty}^{+\infty} f\,\langle\hat{T}_{ab}\rangle_{\omega}\,k^a k^b\,d\lambda + \frac{c}{48\pi}\int_{-\infty}^{+\infty}\left(\frac{df}{d\lambda}\right)^2\frac{1}{f}\,d\lambda
$$

$$
+ \frac{c}{12\pi}\int_{-\infty}^{+\infty}\left(\sigma'^2 f^2 + f'\sigma' f\right)\frac{1}{f}\,d\lambda + \frac{c}{12\pi}\int_{-\infty}^{+\infty} f\left[\sigma'' - \sigma'^2\right]d\lambda
$$

$$
= J_\gamma[g_{ab},k^a,f,\omega]
$$

$$
+ \frac{c}{12\pi}\int_{-\infty}^{+\infty}\left(\sigma'^2 f - f\sigma''\right)d\lambda + \frac{c}{12\pi}\int_{-\infty}^{+\infty} f\left[\sigma'' - \sigma'^2\right]d\lambda
$$

$$
= \boxed{J_\gamma[g_{ab},k^a,f,\omega]}.
$$

(111)

We arrived from the third to the fourth line using integration by parts. The final piece of the proof, is to show that $J_\gamma[g_{ab},k^a,f,\omega]\geq 0$. This result has been proven in [8]. $\qquad\square$

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
