# Peer review of "The Return of the Singularities: Applications of the Smeared Null Energy Condition"

_SciPost Physics, doi:SciPost Phys. 13, 001 (2022)_

## Round 1 · Referee Report · Anonymous · 2021-7-17

Report
Authors study a recently proposed smeared null-energy condition(SNEC). This can be understood
as a bridge between null-energy condition(NEC) and averaged null-energy condition(ANEC), as their
bound takes into account a part of a null geodesic. This is an interesting proposal and the paper
gives enough motivation and examples to be published. Moreover, in Section 2.2 they address existing criticism regarding the feasibility of SNEC.
However, I would like the authors to address a few questions:
# Are there other ways to impose Lorentz-invariant cut-off in Section 2.2? For example using
dimension regularization?
# Why does $\Delta \rho_{OUT}$ include the integral from $\rho_0/2$ to infinity? As far as
I understand the construction of ref. [6], $AdS_2$ within $[-\rho_0/2,\rho_0/2]$ is glued directly
to the outside black hole geometry.
# Also as far as I understand, wormhole length $l$ is not a free parameter, as mentioned after
eq. (45), but is fixed by patching the metric to the outside geometry.
# In the beginning of the paper it is presented as if the function $g(\lambda)$ can be
almost anything. However later in Sections 3.3.1 and 3.3.2 authors concentrate on Gaussian.
Is it done for illustration purposes or the "smearing function" actually have to have a "bump" form,
like a Gaussian or Lorentz?
I think readers would benefit a lot if all the requirements on $g$ are listed in the Introduction.
Looking at eq. (57) it seems relatively easy to prove this bound for any $g$,
since Euler--Lagrange extremal equations will actually give a minimum.
Also I noticed a few typos:
# Subscript "uv" in eq. (24) and right above it.
# "g" right after eq. (56).
# "teh" right after eq. (52).
# "Black" right after eq. (44).
# "Equation eq." right before eq. (65).
Author: Eleni-Alexandra Kontou on 2021-11-09 [id 1929]
(in reply to Report 3 on 2021-07-31)1) By semi-classical gravity, we mean a treatment of classical gravity coupled to quantum fields, where the metric is sourced by the expectation value of the stress tensor. This description can certainly be valid in quantum states with more than two particles, but it breaks down when the geometry is highly curved, or when the stress tensor has large quantum fluctuations on the characteristic distance scales relevant to the solution. A detailed understanding of the regime of validity of the semi-classical approximation is an interesting question, but not the focus of this work. We have added a few sentences clarifying this. 2) We have added some more details about that issue. 3) The null vector that the stress-energy tensor is contracted with is the tangent vector to the geodesic. Thus, the condition is invariant under reparametrization. We have added a comment about that. 4) (32) is not a required condition, it is assumed there for simplicity. The letter f appears twice in this section accidentally instead of g, we have fixed those typos. 5) B is theory independent for free fields. Its comparison with 1/q in 3.3.1 is simply a comparison of dimensionless numbers not an indication that B depends on q. We have rephrased that part to make it clearer. 6) We thank the referee for pointing out these references which we now cite. Indeed this could be interesting to look at for future work. 7) We have added a reference to the appendix. 8) Agree, it has been fixed. 9), 10) and 11): There is a frame independent way to impose the cutoff which we discuss in pg. 10. For this cutoff it is not necessary to pick a null ray. However, we have kept the cutoff on transverse momenta as this construction is connected to the proof of the bound using pencils. 12) We have added more details about the pencil construction including a figure. 13) We have added a reminder. 14) Yes, this is correct. We have added a clarification to the argument. 15) See point 5). 16) We have added a sentence about this for clarity. 17) There is a relevant comment in that section: “We should note that while AdS2 is not globally conformal to Minkowski the bound can be used approximately in the wormhole case as long as we stay away from the boundaries.” 18) Previously we showed that for Q_1=1 to obtain the optimal result for the location of the hypersurface (as close as possible to the classical horizon), the NEC has to be violated for R_s/3. So the requirement is that the NEC is violated for a distance of order of the Schwarzschild radius. We have added a footnote to remind the reader. 19) That is an interesting question which we intend to investigate in future work.

---

## Round 1 · Referee Report · Anonymous · 2021-7-18

Report
One of the most important unsolved questions in general relativity is how conditions on the stress-energy tensor restrict the existence of "exotic phenomena," such as closed timelike curves or traversable, wormholes, in the resulting spacetime geometry. Once quantum effects are included, it appears necessary that such "energy conditions" hold only as spacetime averages rather than pointwise. This paper explores applications of a recently proposed condition, the Smeared Null Energy Condition (SNEC). As the authors describe, this condition has both strengths and weaknesses in comparison to other proposed conditions, so while I find it unlikely to be the last word in the subject, I think it is a helpful addition to the field, especially in the context of the examples studied here. As a result, I would recommend that the paper be accepted for publication, provided that the follwing (related) points can be addressed or clarified. Most of these questions arise already in the Introduction, though their effects show up in later sections; while I recognize that the SNEC has been introduced in a previous paper, it would still help to review it more precisely here.
1) The SNEC is defined in terms of a null geodesic with "length" tau. This quantity cannot refer to proper time for a null geodesic, so it must instead represent an affine length along the geodesic. Since this choice is arbitrary, it should be demonstrated that the condition is invariant under reparameterization.
2) It is not clear whether a normalization condition on the smearing function g(lambda), such as that used in Sec. 3.1, is in general assumed or required.
3) In Eq. (1) T_{kk} is defined to represent the stress-energy tensor contracted with an arbitrary null vector. In Eq. (2) the same symbol is used, but it is not made clear whether the contraction is now with an arbitrary null vector or the tangent vector to the geodesic. (And similarly in later uses of this expression.)
4) The nature of the "constant" B should be clarified. Is it simply a pure number like 1/(32 pi)? If not, does it depend on the matter content and coupling constants of the theory (including G_N), the spacetime dimension, the UV cutoff, the smearing function, the choice of affine parameter, etc.? It is also not clear to me why the factor of 4 is included explicitly in the definition rather than being absorbed into B.
5) In Sec. 2.1, "pencils" should be defined.
6) The first sentence of Sec. 2.3 seems garbled.
7) In Sec. 3.1 there are several uses of the function f, which is undefined and seems to be referring to the function g.
Author: Eleni-Alexandra Kontou on 2021-11-09 [id 1928]
(in reply to Report 2 on 2021-07-18)
1) SNEC is invariant under reparametrization of the affine parameter. We have added a comment about that. 2) That condition is not generally required and it is only assumed in certain examples. 3) The null vector is indeed the tangent vector to the geodesic. We have added a comment about this fact as it was not clear. 4) Subsection 2.2 is devoted to the physical meaning of the dimensionless constant B. Specifically, for free fields in Minkowski spacetime “The constant B in this case arises from the relation between the UV cutoff and the Planck length”. We conjecture that a SNEC-like inequality will be valid in situations with curvature and interactions. In those cases B would have a different value that is currently unknown. The factor 4 is due to the connection of Eq.(3) with the form of SNEC in Ref.[4] where B was first introduced. We have now made that connection clear. 5) The pencils are essentially a null line regularized by a (d-2) transversal area which makes them 2+1 d. But effectively they are only 1 d. We have added that comment to the manuscript and a figure for clarity. 6) We have rephrased that sentence as it could be confusing. 7) The letter f appeared twice in this section accidentally instead of g, we have now fixed those typos.

---

## Round 1 · Referee Report · Anonymous · 2021-7-31

Report
This interesting paper motivates a smeared null energy condition in semi-classical gravity previously conjectures by two of the authors, checks the validity of the bound in various examples including a purported counterexample, and derives a singularity theorem assuming the bound. It is of great interest to establish bounds on the stress energy tensor in gravity given that quantum effects violate the classical null energy conditions. The bound discussed here applies to finite length segments of null geodesics, and thus can be used in spacetimes such as evaporating black holes and traversable wormholes that lack relevant achronal geodesics. Moreover, it does not entanglement entropies or other non-linear properties of the state, and thus is complementary to bounds like the quantum null energy condition.
I recommend this paper be published after some clarifications detailed below.
Requested changes
1. What is the exact definition of the semi-classical regime in
which the authors expect their bound to be valid? Much of the
discussion seems to be in the context of 1-loop corrections to the
matter stress energy tensor. What if 1-loop metric corrections are
included, or if matter states beyond squeezed states (that is
constructed by applying more than 2 creation operators, so roughly 2 loop or higher corrections) are considered.
2. It would be helpful for the reader to review the SNEC a little more extensively. In particular, the appearance of the Newton
constant as a proxy for the field theory UV cutoff seems important to explain near equation (3).
3. The SNEC doesn't appear to be invariant under changes of the
affine coordinate on the null geodesic. In particular, the LHS and
RHS of (3) appear to scale oppositely under rescaling of lambda.
Can't one use this to make the RHS arbitrarily small? Perhaps there is a typo or a condition on lambda.
4. In section 3.1, the above appears to be addressed by requiring
(32). If that is so, this should be included with (3). The function
f after (31) is not defined. Does it refer to g?
5. It is not clear whether B is intended to be a theory independent
number, or a property of a given QFT. In section 2 it seems that it
is meant to be theory independent, given that the number of fields is treated separately. At the end of section 3.3.1, it however
appears to be suggested that B could be of order 1/q, which is
theory dependent. Can the authors clarify the remark there? In the summary on page 22, the result of 3.3.1 is described as the bound not being saturated. So perhaps the end of 3.3.1 should be reworded.
6. Do the authors have any thoughts on how the argument of 2.1 for free theories could be extended to QFTs that are interesting CFTs in the UV? Then the structure of the theory on the light sheet is different, for example as studied in arxiv:1406.4545 or in the CFT proofs of the ANEC in arxiv:1610.05308. Incidentally, the authors might consider citing that work along with the entanglement based proofs.
7. It would be helpful to the reader to point in section 2.1 to the nice proof of (9) given in the Appendix A.
8. "that" should be "than" at the bottom of page 4.
9. I did not understand the reasoning on page 8. The domain of validity of semi-classical gravity is a property of the
configuration, not the choice of null geodesic. If the state were a
plane wave with momentum k, the only invariant measure of breakdown of the semi-classical theory would be the invariant mass. It shouldn't matter whether the component traverse to a particular null ray is large.
10. A related question. Is it Lorentz invariant to demand an energy
cutoff on the pencils defined with respect to a particular null directon?
11. The argument on page 9 appears sufficient, and does refer to an intrinsic property of the state, so the previous argument seems superfluous.
12. It would benefit the reader to include a little more review of
the concept of pencils.
13. It would be helpful to remind the reader the origin of the maximum value of d quoted on page 12.
14. I found the paragraphs at the end of 3.3.1 a little confusing.
Since sigma has a maximum value of 4.13, why is the limit in (65)
relevant? The sentence on page 14 including "it is pretty clear that the small sigma case..." is hard to understand. I think the point is that for small sigma, the bound is clearly obeyed, the most interesting constraint is for the maximum sigma. For the latter case, the bound is also not saturated, due to the factor of q. Is that correct?
15. In 4d language, q is a paramter of the configuration, not the
theory, since it is the effective number of 2d fields. So how could
B depend on q? I'm puzzled by the last sentence of 3.3.1.
16. I believe that in 3.3.2 the authors mean to analyze the AdS2 x
S^2 region of the 4d wormhole in the language of dimensional
reduction to 2d, in particular, in terms of the 2d Newton's constant in their bound. If that is so, it would be good to state it more clearly.
17. However, the resulting 2d theory will not in general be a CFT,
so the language used in 3.3.2 seems not quite precise.
18. The discussion at the bottom of page 18 was a little unclear to
me. Why is it important that the NEC be violated over a length of
order the Schwarzschild radius?
19. In the context of traversable wormholes, for example of the type discussed in 3.3, can one derive a general bound on the length of the achronal segment and its distance from the AdS2 x S^2 throat region, in a model independent way, using the SNEC? For example, in situations with less symmetry.
Author: Eleni-Alexandra Kontou on 2021-11-09 [id 1927]
(in reply to Report 3 on 2021-07-31)
Reply to report 1:
1) We thank the referee for this suggestion however we are unsure on how to impose a Lorentz invariant cutoff using dimensional regularization. Our way of imposing the cutoff is certainly not unique. 2) As far as we understand the construction, we think our limits of integration are correct. The variable \rho goes from -infinity to infinity in the throat. 3) The referee is correct that the l is fixed for each solution. But considering all solutions, it can vary and so in that sense it can be thought of as a free parameter. 4) To have a finite bound it is sufficient that g is differentiable. We now state that clearly at the beginning of the manuscript. Thus the use of the Gaussian is for illustrative purposes. The referee is correct that one can find an optimal g from Eq.(57). This calculation is actually something we did while working on this part. However, we ultimately decided to omit it as it was quite complex and it didn’t add to the main point of this subsection.
Finally, we would like to thank the referee for pointing out the typos.

---

## Editorial Decision

published